# Thermal Degradation Process of Ethinylestradiol—Kinetic Study

Sebastian Simu [1], Adriana Ledeţi [1], Elena-Alina Moacă [2,*], Cornelia Păcurariu [3], Cristina Dehelean [2], Dan Navolan [4] and Ionuţ Ledeţi [1]

1. Advanced Instrumental Screening Center, Faculty of Pharmacy, Victor Babeş University of Medicine and Pharmacy, 2 Eftimie Murgu Square, 300041 Timisoara, Romania; simu.sebastian@umft.ro (S.S.); afulias@umft.ro (A.L.); ionut.ledeti@umft.ro (I.L.)
2. Research Centre for Pharmaco-Toxicological Evaluation, Faculty of Pharmacy, Victor Babeş University of Medicine and Pharmacy, 2 Eftimie Murgu Square, 300041 Timisoara, Romania; cadehelean@umft.ro
3. Faculty of Industrial Chemistry and Environmental Engineering, Politehnica University Timisoara, 2 Victoriei Square, 300006 Timisoara, Romania; cornelia.pacurariu@upt.ro
4. Faculty of Medicine, Victor Babeş University of Medicine and Pharmacy, 2 Eftimie Murgu Square, 300041 Timisoara, Romania; navolan.dan@umft.ro
* Correspondence: alina.moaca@umft.ro

**Abstract:** The present study reports the results obtained after the analysis of the thermal stability and decomposition kinetics of widely used synthetic derivative of estradiol, ethinylestradiol (EE), as a pure active pharmaceutical ingredient. As investigational tools, Fourier transformed infrared spectroscopy (FTIR), thermal analysis, and decomposition kinetics modeling of EE were employed. The kinetic study was realized using three kinetic methods, namely Kissinger, Friedman, and Flynn-Wall-Ozawa. The results of the kinetic study are in good agreement, suggesting that the main decomposition process of EE that takes place in the 175–375 °C temperature range is a single-step process, invariable during the modification of heating rate of the sample.

**Keywords:** ethinylestradiol; isoconversional kinetics; thermal stability; degradation

## 1. Introduction

Ethinylestradiol (17 alpha-ethinylestradiol, abbreviated EE) is a synthetic derivative of estradiol [1] and is an estrogen receptor agonist that binds to both forms of the receptor (ERα and ERβ) and to the G protein coupled estrogen receptor (GPER) [2]. If compared to estradiol, EE is characterized by a higher bioavailability when administered orally, increased resistance to metabolism, and stronger effects in certain parts of the body [1]. EE is commonly used in combination with progestin as a contraceptive, in the treatment of certain gynecological disorders, hormone-dependent cancers (such as prostate and breast cancer), and menopausal symptoms [1]. As a contraceptive, EE is usually available in low doses, in combination with progestin, to reduce the risk of side effects. The intensity of these side effects depends on the dose and route of administration and include weight gain, headaches, nausea, bloating, breast tenderness, and overall feminization (in males) [3]. Long-term administration may lead to an increased risk of blood clots, cardiovascular issues, liver damage, and high doses of EE have been associated with an increased risk of endometrial cancer [1]. More recently, a pilot clinical trial study was carried out by Cortés-Algara et al. regarding the use of EE along with norelgestromin as immunoregulators incorporated in transdermal patches, as an option for the treatment of COVID-19 disease [4]. Moreover, the role of several estrogens, including EE, in menstrual migraine was also studied [5]. The structural formula of EE is presented in Figure 1.

**Figure 1.** Structural formula of EE.

In the pharmaceutical market, worldwide, there are numerous dosage forms containing EE as the solitary active pharmaceutical ingredient (API), or alongside with other API (such as dienogest, levonorgestrel, chlormadinone, norethisterone, norelgestromin and desogestrel), designed for oral (tablets and coated tablets), vaginal (rings), or transdermal (patches with extended release) administration. Dependent of the administration route, the content of EE in each formulation varies commonly between 20 and 50 micrograms per tablet, up to 600 micrograms per patch and 2.7 mg per ring [6]. Since the amount of EE (micrograms) in each formulation is considerably lower than the amount of excipients (milligrams), the development of stable formulations is an imperious demand.

Kinetic analysis—as a direct implementation of thermal analysis—is an important investigational tool for characterization of drugs, including their behavior in pharmaceutical formulations, leading to crucial data regarding lifetime, shelf-life, and degradation mechanism of investigated compounds [7–12]. Taking into account the fact that the mechanism of thermolysis is not known for the majority of organic molecules lead to the main advantage of implementing isoconversional methods, since they allow the estimation of activation energy without knowing or assuming an explicit model for the differential or integral conversion functions [13]. Heterogeneous kinetics has developed over the years and during this period of time, a series of recommendations was elaborated by the International Confederation for Thermal Analysis and Calorimetry (ICTAC) Kinetics Committee, in order to improve the quality of these studies [14]. Numerous papers presents, also, the advantages and disadvantages of each type of kinetic modeling approaches, including model-fitting, model-free (isoconversional), and deconvolution analyses [15–19].

The kinetic method proposed by Kissinger and published initially in 1956 [20] and later in 1957 [21] is one of the most popular kinetic protocol that allows the estimation of the activation energy by using differential scanning calorimetry (DSC) data, differential thermal analysis (DTA) data, or derivative thermogravimetry (DTG) data. The simplicity of its use—however, it is tricky for non-expert users since the method can lead to false conclusions mainly for complex degradative processes—reveals a solitary activation energy value, based on the hypothesis of single-step kinetics. For this reason, methods such as Kissinger, classic Ozawa or ASTM E698 are always preliminary to isoconversional studies, which offer by far a more complete and objective perspective over the investigated processes [18].

Regarding the state-of-art for stability and degradation of EE, the literature data reveal several studies carried out between 1993 and 2022, but none of them refer to heterogeneous degradation of this API in solid state. Recent contributions were reported regarding the degradation of EE by different bacterial strains during wastewater treatment, including *Enterobacter tabaci* [22] and *Acinetobacter* [23], and as well the evaluation of kinetics of natural degradation and identification of the resulted products during photodegradation and oxidation were carried out [24]. Moreover, several studies were published regarding the risks and pollution characteristics of contaminants for aquatic ecosystems, including the endocrine disruptors such as EE [25–28]. Adsorption of EE from different aqueous environments was also studied [29–32], and as well of the effect over cellular and molecular variations of microalgae such as *Chlorella pyrenoidosa* [33].

Since the literature data reveal no information regarding the processes of heterogeneous decomposition of this API, we set our goal into carrying out an isoconversional kinetic study according to ICTAC 2000 protocol for this API, under thermal stress [34–37], using the derivative thermogravimetric (DTG) data collected for five different heating rates β = 2, 4, 6, 8, and 10 °C/min. The obtained results were processed according to Kissinger method, the differential method of Friedman, and the integral method of Flynn–Wall–Ozawa.

## 2. Materials and Methods

### 2.1. Samples and Preparation

Ethinylestradiol (EE), a commercial product of Sigma-Aldrich (St. Louise, MO, USA) was used without further purification. The purity of EE was according to European Pharmacopoeia (EP) Reference Standard. Up to the use, the sample was stored according to recommendations made by the supplier.

### 2.2. FTIR Investigations

Fourier-transform infrared spectroscopy studies (FT-IR), were performed using a Shimadzu Prestige-21 spectrometer (Duisburg, Germany), at 24 °C. The operating parameters set were: resolution of 4 $cm^{-1}$ within the spectral range of 400–4000 $cm^{-1}$, using KBr pellets. In the spectroscopic description of bands, the following abbreviations are used for different types of vibration: ν for stretching vibration, δ for internal deformation (bending).

### 2.3. Thermo-Analytical Investigations

The stability of the samples was performed by thermal behavior assess, using a Netzsch STA 449 C instrument (Netzsch-Gerätebau GmbH, Selb, Germany), in the range of 20–500 °C, air atmosphere, at 2, 4, 6, 8, and 10 °C/min heating rates. Each sample was exactly weighed in aluminum crucibles and the analysis was performed under artificial air at a flow rate of 20 mL/min. Air atmosphere was chosen since most of APIs are processed and stored under usual ambient atmosphere. The analysis was carried out in duplicate and the results are practically identical.

### 2.4. Kinetic Study

The kinetic processing of the data (Friedman and Flynn-Wall-Ozawa methods) was carried out on the main decomposition process of EE, using the AKTS—Thermokinetics Software (AKTS AG TechnoArk, Siders, Switzerland). The classical Kissinger method was employed using a template file created by our research team. All the aspects regarding the theoretical foundation and advantages of isoconversional kinetics are presented exhaustively in numerous papers [38–40].

## 3. Results and Discussion

### 3.1. FTIR Investigations

FTIR spectroscopy was used as an investigational technique for characterization of EE. The FTIR spectrum of EE on spectral range 4000–400 $cm^{-1}$ is shown in Figure 2. Literature data present several characteristic bands for infrared spectral investigation of EE, especially for the compound adsorbed from aqueous medium [41]. However, a complete description of FTIR bands is not presented in the literature regarding the spectroscopic analysis of EE.

The stretching vibrations ν(O–H) for both OH moieties from the EE structure are observed in the spectral range 3650−3100 $cm^{-1}$ as a broad band, suggesting the intense H-bonding between the molecules in solid state, overlapped with the sharp band of superficially adsorbed water: the bands are evidenced at 3606.89, 3500.80, and 3292.49 $cm^{-1}$, respectively. The strong band at 3321.42 $cm^{-1}$ can be assigned to ν(≡C–H) from alkyne moiety. The symmetric and asymmetric stretching vibrations for other C–H bonds, namely ν(C–H), including the ones from $CH_3$ moiety and $CH_2$ ones are represented by the bands at 2972.31, 2935.66, and 2866.22 $cm^{-1}$. The sharp, weak bands at 2357.01 and 2322.29 $cm^{-1}$ are characteristic bands, well individualized in the FTIR spectrum of compounds contain-

ing -C≡C- moieties, i.e., the ethinyl moiety of the API. The bands from 1614.42, 1585.49, 1500.62, 1471.69, 1446.61, and 1435.04 cm$^{-1}$ are due to symmetric C-C stretching ($\nu$C-C and $\nu$C=C), as well for $\delta_{as}$(CH$_3$) and $\delta$(CH$_2$). The bands from 1373.32, 1357.89, 1298.09, and 1286.52 cm$^{-1}$ are the consequence of symmetric methyl bending $\delta_s$(CH$_3$) and the latter two for the hydroxyls, $\delta$(COH), respectively. The bands at 1255.66 and 1056.99 cm$^{-1}$ are probably due to $\nu$(C–O) vibration, while the other bands (recorded at 1182.36, 1134.14, 1111.00, 1020.34, 972.12, 929.69, 914.26, 879.54, 819.75, 788.89, 680.87, 644.22, 621.08, 569.00, 524.64, and 441.70 cm$^{-1}$) from the fingerprint region are due to skeleton vibration and different combination bands, that cannot be correctly attributed to a certain bond without carrying theoretical simulations of vibrational spectra using density functional theory [42].

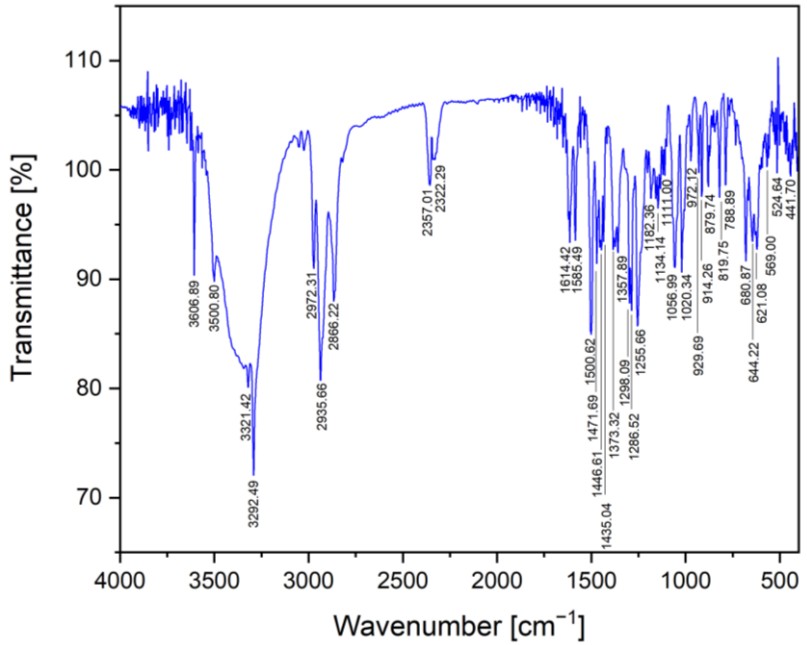

**Figure 2.** FTIR spectrum of EE on spectral range 4000−400 cm$^{-1}$.

### 3.2. Thermoanalytical Investigations

Thermal stability of EE was carried out in dynamic oxidative atmosphere at a heating rate of 2 °C/min, as shown in Figure 3.

According to the thermoanalytical curves depicted in Figure 3, EE is thermally stable up to 71 °C, when a mass loss process begins, taking place between 71 and 101 °C (mass loss 3.18%, DTG process between 69 and 101 °C, DTG$_{peak}$ at 87 °C, DSC process 61 and 101 °C, DSC$_{peak}$ at 88 °C). This first process is attributed to the superficially adsorbed water removal from the sample. Anhydrous EE is stable up to 177 °C, when a decomposition process begins, overlapping the melting of the API (DSC endothermal process between 177–191 °C, DSC$_{peak}$ at 185 °C), in good agreement with the melting interval mentioned on PubChem, suggesting the existence of polymorphic form II [43], instead of polymorphic form I, which melts at 146 °C [44].

The main decomposition process of EE takes place in the temperature range 187–324 °C (mass loss 59.93%, DTG$_{peak}$ at 293 °C), accompanied by an exothermic effect on the DSC curve, with the peak at 287 °C. Further on, with the increase of temperature, the thermoanalytical profile becomes more complex, since the overlapping thermal degradation process takes place. At 500 °C, the residual mass is 14.18%, suggesting an advanced degradation of the structure under thermal stress.

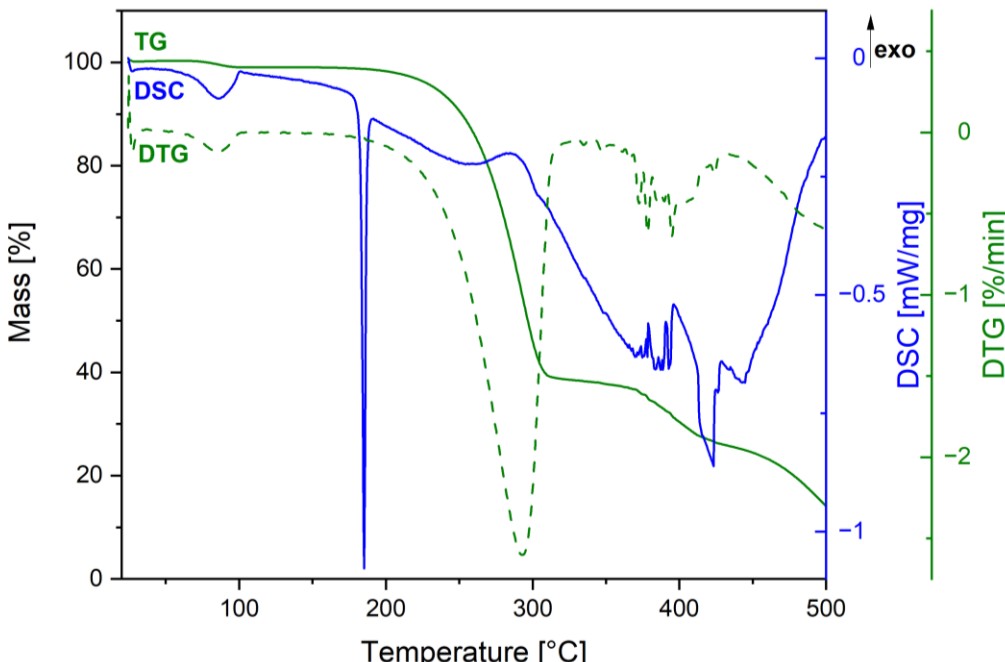

**Figure 3.** Simultaneous−determined TG (thermogravimetric), DTG (derivative thermogravimetric), and DSC (differential scanning calorimetry) curves in air atmosphere at β = 2 °C/min for EE on temperature range 25–500 °C.

### 3.3. Kinetic Study

The kinetic analysis was carried out over processed DTG, obtained in dynamic air atmosphere for the following five heating rates β: 2, 4, 6, 8, and 10 °C/min. The main decomposition process of EE that was subjected to kinetic analysis is the one that takes place after the formation of anhydrous form, which takes place in the following temperature ranges vs. heating rate: β = 2 °C/min (179–330 °C, $T_{max}$ at 293.8 °C), β = 4 °C/min (179–348 °C, $T_{max}$ at 311.7 °C), β = 6 °C/min (179–355 °C, $T_{max}$ at 319.8 °C), β = 8 °C/min (182–377 °C, $T_{max}$ at 326.9 °C), and β = 10 °C/min (187–378 °C, $T_{max}$ at 332.9 °C). A preliminary kinetic study was realized using the Kissinger method, which is based on the assumption that the degree of conversion is a constant and non-dependent of the heating rate at the DTG peak ($T_{max}$) (Equation (1)):

$$\ln \left( \beta \cdot T_{max}^{-2} \right) = \ln \left( A \cdot R \cdot E_a^{-1} \right) + \ln \left[ n \cdot (1 - \alpha_{max})^{n-1} \right] - E_a \cdot R^{-1} \cdot T_{max}^{-1} \quad (1)$$

The estimation of the activation energy of the decomposition ($E_a$) can be made by evaluating the slope of the linear plotting for experiments carried out at the five different heating rates (Figure 4), revealing that for the main stage of EE degradation, the value for $E_a$ is 107.91 kJ/mol.

Even if current protocols regarding the kinetic studies highly recommend the use of isoconversional methods, as requested by international conventions established by the ICTAC protocols, the use of simple methods, such as Kissinger or ASTM E698 is valuable, since it permit a classification of kinetic mechanism for decomposition of organic molecules, and a separation of simple vs. complex processes. It is well-known that organic molecules possess numerous pathways of thermolysis since they contain numerous covalent bonds, and the intensity of thermal stress can determine the modification of the mechanism with increasing of heating rate, and as a consequence, the comparison of the results obtained by classical kinetic methods vs. isoconversional ones can lead to valuable information regarding the complexity of the degradation. Following this hypotheses, if the result obtained by Kissinger method is in good agreement with the ones suggested by differential

and integral isoconversional methods, it can be said that the heating rate has no influence over modification of degradation mechanism.

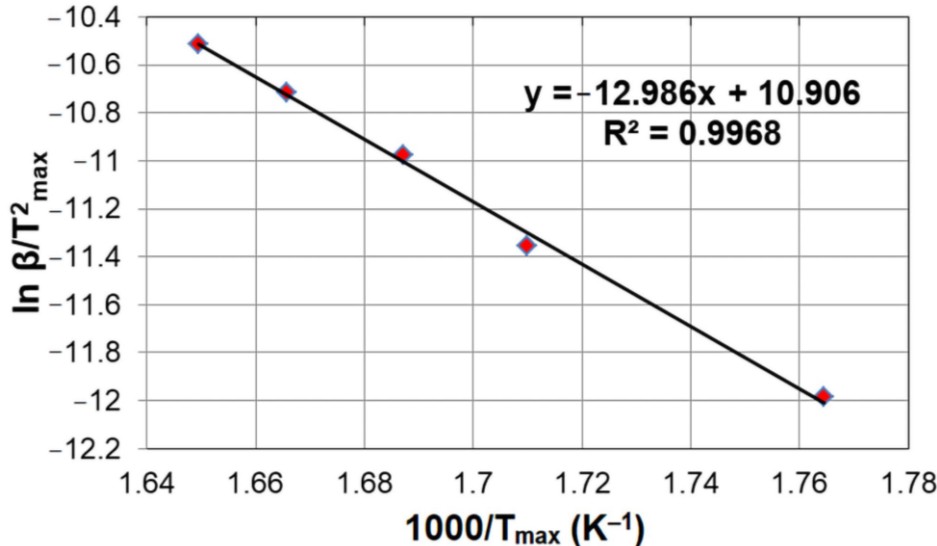

**Figure 4.** Plotting of Kissinger kinetic method for EE.

Two isoconversional methods, namely the differential method of Friedman (Fr) and the integral method of Flynn–Wall–Ozawa (FWO), were used in order to evaluate the values of $E_a$ of the decomposition vs. conversion degree $\alpha$. The reaction progress vs. temperature is presented in Figure 5. As can be shown from Figure 5, the decomposition process is shifted to higher temperatures due to thermal inertia of the sample, as the heating rate increases.

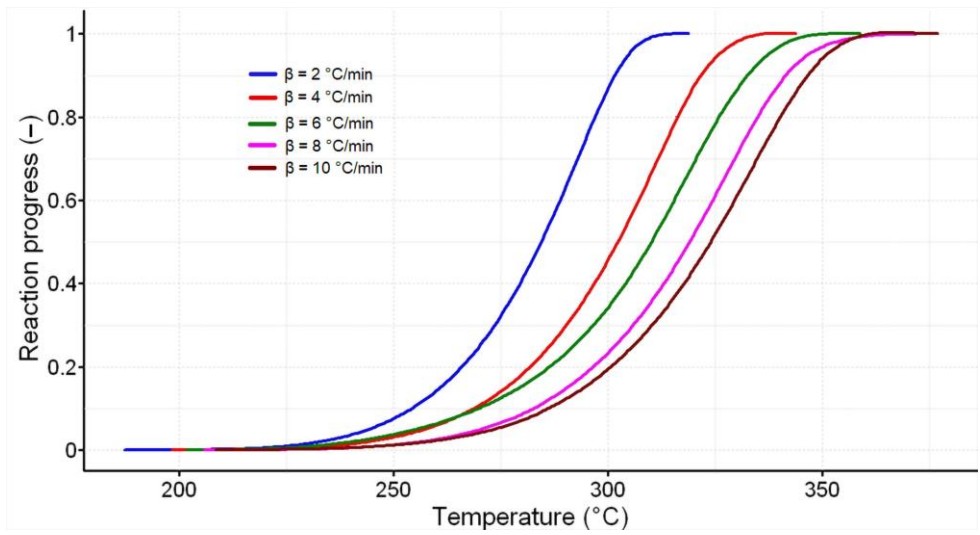

**Figure 5.** The reaction progress vs. temperature dependence for thermal decomposition of EE: curve 1 corresponds to $\beta$ = 2 °C/min, curve 2 to $\beta$ = 4 °C/min, curve 3 to $\beta$ = 6 °C/min, curve 4 to $\beta$ = 8 °C/min and curve 5 to $\beta$ = 10 °C/min, respectively.

The same tendency can be seen in Figure 6, where the plotting of reaction rate vs. temperature reveals the shifting of maximum heating rate at higher temperatures, as the hating rate increases. Moreover, it can be seen that the reaction rate is increased not only by temperature, but also by the heating rate of the sample.

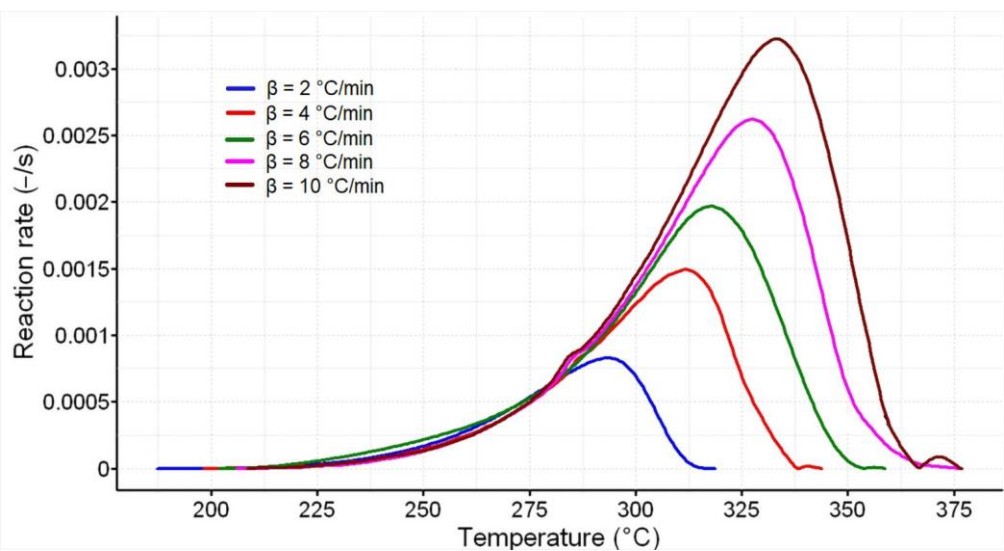

**Figure 6.** The reaction rate vs. temperature dependence for thermal decomposition of EE; curve 1 corresponds to β = 2 °C/min, curve 2 to β = 4 °C/min, curve 3 to β = 6 °C/min, curve 4 to β = 8 °C/min and curve 5 to β = 10 °C/min, respectively.

The theoretical background of isoconversional methods is extensively reported in literature [45–51]; however, for a facile understanding of the results, we briefly present in this paper the final models implemented in kinetic analysis of EE.

The linearized mathematical equation of Friedman method (Fr) [45] is shown in Equation (2).

$$\ln\left(\beta \cdot d\alpha/dT\right) = \ln\left[A \cdot f(\alpha)\right] - E_a \cdot R^{-1} \cdot T^{-1} \tag{2}$$

For selected α at each heating rates, the plot of $\ln\left(\beta \cdot d\alpha/dT\right)$ vs. $1/T$ is linear. By evaluation of the slopes of these graphical representations (Figure 7), the values of the activation energy of the decomposition ($E_a$) are revealed (Table 1).

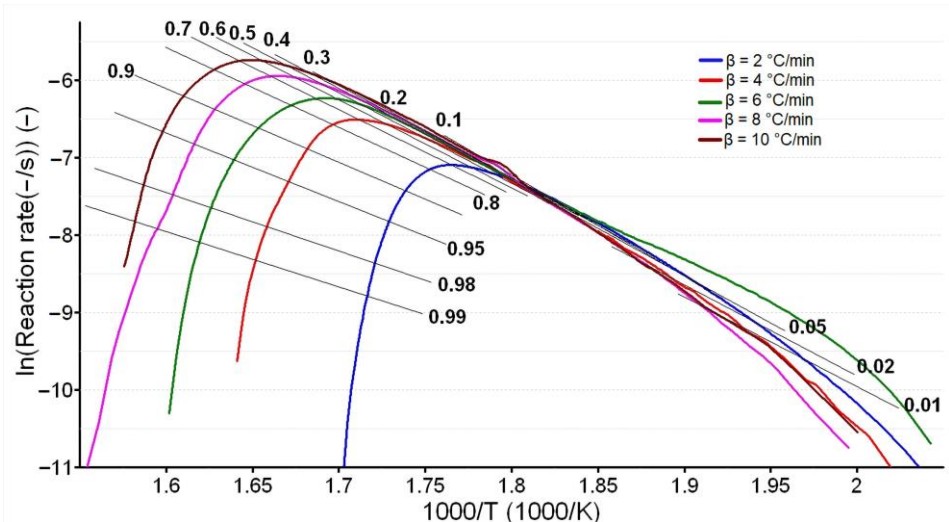

**Figure 7.** Friedman's linear plotting for decomposition of EE.

**Table 1.** Evaluation of the decomposition $E_a$ values vs. conversion degree obtained by the two isoconversional methods and the mean value of $E_a$.

| Conversion Degree | $E_a$ (kJ/mol) vs. $\alpha$ for EE | |
|---|---|---|
| $\alpha$ | Fr | FWO |
| 0.05 | 101.92 | 112.10 |
| 0.10 | 108.76 | 114.01 |
| 0.15 | 103.54 | 114.02 |
| 0.20 | 102.06 | 112.83 |
| 0.25 | 99.95 | 111.48 |
| 0.30 | 99.61 | 110.32 |
| 0.35 | 99.02 | 109.31 |
| 0.40 | 98.52 | 108.49 |
| 0.45 | 97.81 | 107.75 |
| 0.50 | 97.08 | 107.06 |
| 0.55 | 96.17 | 106.41 |
| 0.60 | 94.96 | 105.75 |
| 0.65 | 93.33 | 105.11 |
| 0.70 | 91.06 | 104.38 |
| 0.75 | 88.28 | 103.56 |
| 0.80 | 85.31 | 102.56 |
| 0.85 | 82.16 | 101.32 |
| 0.90 | 79.34 | 99.74 |
| 0.95 | 73.76 | 97.45 |
| $\overline{E_a}$ (kJ/mol) | 94.35 ± 9.00 | 107.03 ± 4.82 |

The Flynn–Wall–Ozawa (FWO) method [48,49] is represented, after Doyle linearization, by the mathematical model presented in Equation (3), where $g(\alpha)$ is the integral conversion function.

$$\ln \beta = \ln \left[ A \cdot E_a \cdot R^{-1} \cdot g^{-1}(\alpha) \right] - 5.331 - 1.052 \cdot E_a \cdot R^{-1} \cdot T^{-1} \tag{3}$$

Similarly, the estimation of activation energy of the decomposition values ($E_a$) for all the conversion degree can be achieved by the plotting of $\ln \beta$ vs. $T^{-1}$, as can be seen in Figure 8 and Table 1, respectively.

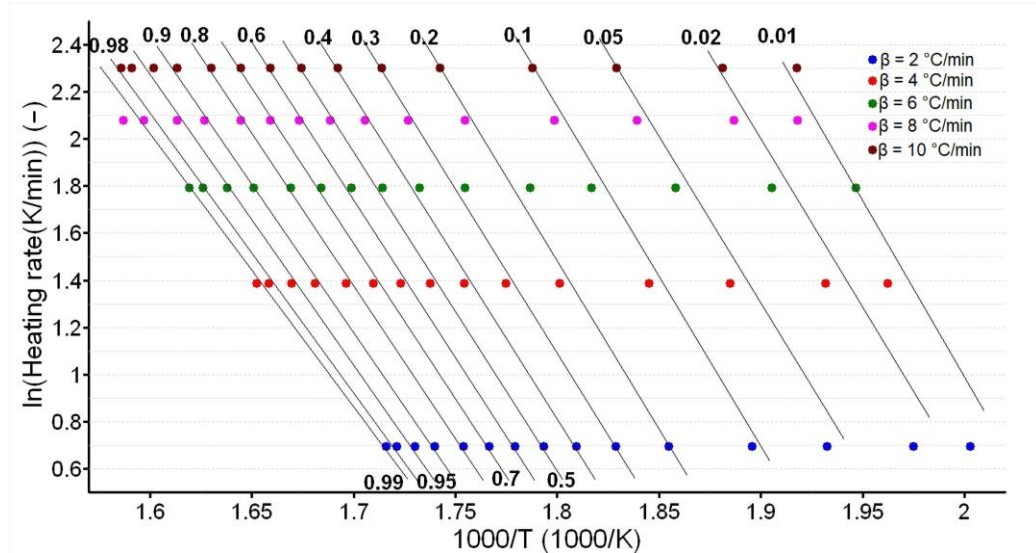

**Figure 8.** Flynn–Wall–Ozawa linear plotting for decomposition of EE.

As can be seen by comparing the results from the Kissinger kinetic method and the two isoconversional ones, the obtained values for the decomposition activation energies

are in good agreement. Moreover, the fitting of $T_{max}$ points from the Kissinger method suggests a very good fitted regression model ($R^2 = 0.9968$), leading to a possible conclusion that the degradative mechanism of EE under thermal stress is independent of the heating rate. However, this conclusion can be sustained only by the correlation of the result with the ones obtained by the use of the two isoconversional methods, the differential and the integral one, since there are situations where the Kissinger plot is almost perfectly linear, without detecting the complexity of the decomposition process(es) [18].

The necessity of estimating the value of activation energy for small variation of conversion degree (5%) is based on the fact that the individual $E_a$ values falling outside the $\pm 10\%$ interval around the medium value, clearly indicate a multi-step degradation process. Otherwise, if the variation of $E_a$ vs. $\alpha$ values are inside the interval $\overline{E_a} \pm 0.1\overline{E_a}$, the mechanism of degradation consists in a single-step process, invariable with the modification of heating rate of the sample.

The isoconversional method of Friedman reveals a variation outside the $\pm10\%$ limit, outside the medium value of $E_a$ for EE solely at conversions superior to 85%, as seen in Table 1. This variation is not observed in the case of the integral method of Flynn–Wall–Ozawa, even if the individual $E_a$ values have the tendency to decrease with the advance of the reaction.

## 4. Conclusions

In this paper, the results obtained after studying the process of heterogeneous degradation of the estrogen medication ethinylestradiol were reported, by means of thermal analysis and isoconversional kinetics. The analysis was also completed by FTIR spectroscopy, in order to prove the identity and purity of the compound.

Thermal analysis revealed that anhydrous EE has a good thermal stability (up to 177 °C), this fact being explained by the existence of the stable steroid moiety in the structure of this drug. The kinetic study was realized using three kinetic methods, namely Kissinger, Friedman and Flynn–Wall–Ozawa. The results of the kinetic study are in good agreement, suggesting that the main decomposition process of EE that take place in the 175–375 °C temperature range is a single-step process, invariable with the modification of heating rate of the sample.

However, due to differential process of the data according to Friedman method, at higher conversion degrees ($\alpha > 0.85$), a clear indication for a modification of the degradative mechanism is revealed. This last observation can be explained by the fact that the main degradation process is immediately followed by another process, more complex, as can be also seen on the thermoanalytical profile of EE. Moreover, the complexity of the second degradative process can be explained by the fact that at higher temperatures, the thermolysis of steroid moiety is accentuated and also overlaps the degradation of the products resulted from the main process, that was kinetically investigated in this study.

**Author Contributions:** Conceptualization, S.S. and I.L.; methodology, A.L., E.-A.M., C.P. and I.L.; software, A.L., E.-A.M., C.P. and I.L.; validation, A.L., C.D., D.N. and I.L.; formal analysis, A.L., E.-A.M., C.P. and I.L.; investigation, S.S., A.L., E.-A.M., C.P. and I.L.; resources, C.D. and D.N.; data curation, S.S., A.L., E.-A.M., C.P. and I.L.; writing—original draft preparation, S.S. and I.L.; writing—review and editing, C.D., D.N. and I.L.; visualization, C.D., D.N. and I.L.; supervision, A.L. and I.L.; project administration, S.S. All authors have read and agreed to the published version of the manuscript.

**Funding:** This research received no external funding.

**Institutional Review Board:** Not applicable.

**Informed Consent Statement:** Not applicable.

**Data Availability Statement:** Authors can provide raw data under request.

**Conflicts of Interest:** The authors declare no conflict of interest.

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
