# Peer review of "Thermal Degradation Process of Ethinylestradiol—Kinetic Study"

_processes, doi:10.3390/pr10081518_

Round 1
Reviewer 1 Report
Comments to Simu et al.
Summary
The manuscript deals with the degradation kinetics of Ethinylestradiol employing Fourier transformed infrared spectroscopy to obtain information of the structure of the substance, thermo analysis to investigate the thermal stability and kinetic modeling to estimate the activation energy of the thermal degradation. As different kinetic methods render similar estimates of the activation energy the authors conclude that the thermal decomposition essentially is a single-step process.
General comments
The topic of the study, concerning the pharmaceutical field, is well within the scope of the Processes Journal. Furthermore, the results, albeit rather theoretical, also have some relevance. In addition, the manuscript contains the customary sections of a scientific article. As for English language, the quality is reasonable; there are no grave grammatical errors, but there is room for improvement in the fluency of the text.
The authors give a sufficient background of the topic in the Introduction section, whereas the description of the experiments in the Materials and Methods section is somewhat superficial. For the reader to be able to repeat the experiments, it would be good to have a more thorough account for the preparation of the samples. Furthermore, the Results section would benefit from some estimates of more tangible nature. Is it for example, from the kinetic results, possible to estimate how the decomposition proceeds as a function of time? In other words, will the degradation of the active substance of the pharmaceutics products become an essential problem at customary temperatures and relevant storage times? In addition, the manuscript presents the numerical results in a way that does not allow the reader to check the calculations. It would be helpful with the raw data in a separate file of supportive information or alternatively in an external repository (e.g. Zenodo).
Specific comments
The first sentence of the abstract is a bit clumsy. The present study reports…
Line 22: The decomposition kinetics in itself is not a tool, rather the decomposition kinetics modeling.
Line 25: takes place
Line 26: Rather invariable during, but what exactly is it that is invariable?
Line 111: Tautology (presentation presented).
Figure 4: How do you arrive at the negative values on the vertical axis? After all, the heating rates, in degrees per minute, are all greater than one and hence their logarithms should be positive.
Figures 5&6: The font of the legends is too small.
Equation (2): The two -1:s should be exponents.
Lines 191-192: heating rate
Equation (3): This is the Doyle linearization of the Ozawa-Flynn-Wall model.
Line 229: The use of the noun fall with the of genitive seems a bit awkward. Maybe case or occurrence would be better. Another alternative would be Ea values falling outside the ±10% interval.

Reviewer 2 Report
The study presents the thermal stability and decomposition kinetics of the widely used synthetic derivative of estradiol. In the manuscript, chemical and thermal analyses were performed. The manuscript is interesting and well written. However, there are several points that I would like to address:
Introduction: a deeper explanation of the impact of the significance of thermal studies on the described effects (lines 56-59) is advisable.
Introduction: other methods of kinetics analysis might be mentioned, i.e. Kissinger method.
Methods: on what basis was the temperature range selected for the thermal tests,
Methods: please precise how many repetitions were performed during experimental studies.
Methods: why thermal tests were performed in the air?
Results: it would be more clear to specify what the determined activation energy refers to, e.g. “activation energy of the decomposition”.
Results: what practical significance does the observation have “Also, the fitting of Tmax points from the ASTM E698 method suggests a very good 225 fitted regression line (R2 = 0.99), leading to the conclusion that the degradative mechanism 226 of EE under thermal stress is independent of the heating rate”
Conclusions: please specify the practical significance of the experiments carried out from the medical poitn of view.
Round 2
Reviewer 2 Report
All my comments were improved in a revised version of the manuscript. I endorse it for publication.